# Regulation of a Novel Splice Variant of Early Growth Response 4 (EGR4-S) by HER+ Signalling and HSF1 in Breast Cancer

**DOI:** 10.3390/cancers14061567

**Published:** 2022-03-18

**Authors:** Jeremy M. Drake, Benjamin J. Lang, Martin Eduardo Guerrero-Gimenez, Jack Bolton, Christopher A. Dow, Stuart K. Calderwood, John T. Price, Chau H. Nguyen

**Affiliations:** 1ProMetTre Cancer Research, Melbourne 3205, Australia; 2College of Health and Biomedicine, Victoria University, Melbourne 8001, Australia; jack.bolton86@gmail.com (J.B.); john.price@vu.edu.au (J.T.P.); chau.nguyen@icmp.int (C.H.N.); 3Department of Radiation Oncology, Beth Israel Deaconess Medical Center, Harvard Medical School, Boston, MA 02215, USA; b.j.lang617@gmail.com (B.J.L.); scalderw@bidmc.harvard.edu (S.K.C.); 4Laboratory of Oncology, Institute of Medicine and Experimental Biology of Cuyo (IMBECU), National Scientific and Technical Research Council (CONICET), Mendoza 5500, Argentina; martine.guerrero@iqvia.com; 5Dorevitch Pathology, Western Hospital, Melbourne 3011, Australia; chris.dow@dorevitch.com.au; 6Department of Medicine, University of Melbourne, Melbourne 3052, Australia; 7Institute for Health and Sport, Victoria University, Melbourne 8001, Australia; 8Australian Institute for Musculoskeletal Science (AIMSS), Victoria University and Western Health, Melbourne 8001, Australia; 9Department of Biochemistry and Molecular Biology, Monash University, Clayton 3800, Australia

**Keywords:** EGR4, HSF1, HER2, breast cancer, molecular stress

## Abstract

**Simple Summary:**

EGR4 is known to play an important role in the proliferation of small cell lung cancer. Our research identified a new, shortened version of this protein (which we named EGR4-S), found in breast cancer but not detectable in normal breast tissue. Interestingly, our findings show that the EGR4-S expressed by breast cancer cells could be reduced by treating the cells with certain targeted cancer therapeutics. However, sustained, high-dose treatment led to EGR4-S becoming less responsive. In addition, we identified an inverse relationship between EGR4-S and molecular stress. When cancer cells were in conditions of increased molecular stress, reduced EGR4-S levels were associated with lower growth rate but enhanced properties associated with higher metastatic potential. Taken together, our research suggests further investigation of EGR4-S is warranted in order to determine its potential as a biomarker for differentiating tumours from normal tissue at the molecular level, as well as its possible resistance to targeted therapies.

**Abstract:**

The zinc finger transcription factor EGR4 has previously been identified as having a critical role in the proliferation of small cell lung cancer. Here, we have identified a novel, shortened splice variant of this transcription factor (EGR4-S) that is regulated by Heat Shock Factor-1 (HSF1). Our findings demonstrate that the shortened variant (EGR4-S) is upregulated with high EGFR, HER2, and H-Ras^v12^-expressing breast cell lines, and its expression is inhibited in response to HER pathway inhibitors. Protein and mRNA analyses of HER2+ human breast tumours indicated the novel EGR4-S splice variant to be preferentially expressed in tumour tissue and not detectable in patient-matched normal tissue. Knockdown of EGR4-S in the HER2-amplified breast cancer cell line SKBR3 reduced cell growth, suggesting that EGR4-S supports the growth of HER2+ tumour cells. In addition to chemical inhibitors of the HER2 pathway, EGR4-S expression was also found to be suppressed by chemical stressors and the overexpression of HSF1. Under these conditions, reduced EGR4-S levels were associated with the observed lower cell growth rate, but the augmentation of properties associated with higher metastatic potential. Taken together, these findings identify EGR4-S as a potential biomarker for HER2 pathway activation in human tumours that is regulated by HSF1.

## 1. Introduction

Cancer is projected to be the most important barrier to increasing life expectancy in the 21st century [1]. In 2019, breast cancer became the most diagnosed cancer in the world (excluding basal cell carcinoma), surpassing lung cancer for the first time [2]. A frequently altered cell signalling pathway in both breast and lung cancer is the Human Epidermal growth factor Receptor (HER) pathway. This HER pathway increases cell proliferation, and changes to components of the HER signalling pathway have been recorded in many types of cancer. For example: approximately 15–25% of breast cancers have an altered (HER2-activated or HER2-enriched) pathway [3,4,5]. Additionally, up to 60% of lung cancers have also been observed to have an altered (HER1-activated) pathway [6,7,8].

Historically, the treatment of solid cancers has involved surgery, radiotherapy, and/or chemotherapy. More recently, as researchers have come to understand the cellular changes that promote cancer growth, targeted therapies that specifically target and arrest these cancer-promoting cellular changes have been developed. Since components of the HER pathway are so frequently altered in cancer cells, many HER-targeted drug therapies have been developed to improve patient prognosis. One class of these drugs, Tyrosine Kinase Inhibitors (TKIs), are effective in treating cancers of the lung and breast that have activated HER signalling pathways. TKI drugs such as lapatinib (in breast cancers), gefitinib, and erlotinib (in lung cancers) are now incorporated into the treatment regimes of people with advanced HER+ breast cancers or HER+ lung cancers because of the prognostic benefits [7,9,10,11].

Despite the improvements in survival time provided by anti-HER therapies in patients with advanced disease, most develop a progressive disease, and mortality rates remain high [11]. Similar to many cancer therapies, the use of these HER pathway-targeted drugs is limited by the problem of resistance—some HER+ individuals are not responsive to these agents (de novo or intrinsic resistance), and others develop resistance to them over time following exposure (acquired resistance) [7]. Several underlying cellular mechanisms have been linked to developing resistance, including an impaired drug binding to the HER receptor and the constitutive activation of signal pathways downstream or parallel to HER. Selecting patients for targeted therapy is therefore becoming reliant on the use of predictive and prognostic biomarkers [12,13].

Cellular stress has long been associated with drug resistance in many types of cancer [14,15,16]. One response to cellular stress, called the Heat Shock Response, is triggered by many forms of cell stress and results in an accumulation of Heat Shock proteins (HSPs) within in the cell [17,18]. The ‘master regulator’ of the Heat Shock Response is the transcription factor Heat Shock Factor 1 (HSF1), which is activated when a cell is under stress [18,19]. HSF1 has long been reported to play crucial roles in cancer progression, metastasis, and drug resistance [19,20].

Our previously published cell model [21] investigated the effects of HSF1 on oncogenicity in human MCF10A mammary epithelial cells. From this research, the Early Growth Response 4 (*EGR4*) gene was identified as being differentially regulated by HSF1 in non-transformed versus H-Ras^V12^ oncogenically transformed breast cells. Previous research on *EGR4* in cancer suggests it may be connected to advanced tumourigenesis, with a role in the cell proliferation and bone metastasis of small cell lung cancer [22,23], as well as an upregulated expression correlating with the breast cancer grade [24]. Given that HSF1 plays a role in cancer progression, metastasis, and drug resistance, and our previous finding that *EGR4* is regulated by HSF1 in breast cancer cells, the aim of this study was to more fully investigate the potential oncogene *EGR4* and its regulation by HSF1.

## 2. Materials and Methods

### 2.1. Generation and Sources of Plasmid Constructs

Plasmid constructs were generated, as described previously [21]. All expression vector sequences were confirmed by DNA sequencing (Micromon DNA Sequencing Facility, Monash University, Melbourne, Australia).

### 2.2. Cell Lines and Cell Cultures

The MCF10A cell line was obtained from the A.T.C.C. (Manassas, VA, USA) and cultured as described previously [25]. T47D cells, SKBR3 cells, Hs578T, and HEK-293T cells were cultured, as described by [21]. MDA-361, ZR75-1, BT-474, MDA-453, MDA-468, MDA-231, BT549, and MCF7 cells were cultured in Dulbecco’s Modified Eagle’s medium supplemented with 10% (*v*/*v*) fetal bovine serum and 1% (*w*/*v*) penicillin/streptomycin. In addition, MCF7 growth media was supplemented with 10 ng/mL insulin. All stable cell lines were generated by retroviral or lentiviral transduction as per [25], with the selection of cells based on GFP described previously [21] to avoid any unnecessary stress stimulus to the cells. Viral stocks were generated by transient transfection of appropriate viral packaging vectors into the HEK-293T cell line, as described by [26]. Drugs/compounds used for treating the cells were purchased from commercial sources, including lapatinib (catalogue number SML2259), erlotinib (catalogue number CDS022564), gefinitib (catalogue number CDS022106), sulforaphane (catalogue number S4441) from Sigma (Saint Louis, MO, USA), and NVP-AUY922 (catalogue number sc-364551) from Santa Cruz Biotechnology (Santa Cruz, CA, USA). Chronic lapatinib treatment of breast cancer cells was performed by growing cells in media supplemented with lapatinib over the course of 7 weeks. The supplemented media was changed every 3 days to maintain living cells and remove dead cells. The lapatinib treatment concentration was doubled at the conclusion of each week for the duration of the experiment. HSF1-targeted shRNAmir (microRNA-adapted short hairpin RNA) vectors were constructed, as described previously [26]. The *EGR4*-targeted shRNAmir vectors were purchased from Millenium Science (Mulgrave, Australia).

### 2.3. Analysis of Cancer Cell Glycolytic Status

Experiments for measuring cellular glycolysis were performed using a Seahorse XF24 Analyser (Agilent, Santa Clara, CA, USA) and Glycolysis Stress Test kit (Agilent, Santa Clara, CA, USA). The optimal cell seeding number was determined by initial titration experiments. The Seahorse sensor cartridge was hydrated for 24 h at 37 °C in XF Calibrant liquid prior to each experiment. Immediately before the experiment, each plate well was loaded with 80 µL of the supplied Glycolysis kit compounds (10 mM glucose, 2.5 µM oligomycin, and 100 mM 2-Deoxy-D-Glucose), according to the manufacturer’s specifications. After seeding 80,000 cells/well, the cells were allowed to adhere to the plate for 3 h before loading both the sensor plate and the cell culture plate into the Seahorse machine for analysis. The conditions comprised of an initial equilibration step (3 cycles of: 3 min—mix, 2 min—wait, 3 min—measure), the addition of a glucose step (3 cycles of: 3 min—mix, 2 min—wait, 3 min—measure), the addition of an oligomycin step (3 cycles of: 3 min—mix, 2 min—wait, 3 min—measure), and finally, the addition of 2-DG (3 cycles of: 3 min—mix, 2 min—wait, 3 min—measure). At the completion of the experiment, the data from each sample was analysed using the “Wave v2.3” software.

### 2.4. Protein Extraction from Cells and Tissues

The generation of protein lysates from cells was performed according to a previously published protocol [27]. Protein lysates from frozen human tissue samples were prepared by crushing the samples using liquid nitrogen and a sterile mortar and pestle, then placing the ground tissue into a homogenisation tube along with 600 µL of PARIS buffer (Thermofisher Scientific, Waltham, MA, USA), 6 µL protease inhibitors, and 6 µL phosphatase inhibitors. Tissue homogenates were then mixed by passing them through a sterile 18 gauge syringe several times before use.

### 2.5. Western Blot Analysis and Antibodies

Equal concentrations of protein lysates were combined with a loading buffer (Invitrogen) and a sample reducing agent (Invitrogen, Carlsbad, CA, USA), denatured for 5 min at 95 °C, then separated through gel electrophoresis in a 4-to-12% Bis-Tris NuPAGE gel (Invitrogen) with 1xMES buffer (Invitrogen) and antioxidant (Invitrogen). Following separation, the proteins were electrophoretically transferred onto nitrocellulose membranes using the iBlot western detection stack/iBlot dry blotting system (Thermofisher Scientific) for seven minutes at 23 V. All membranes were then processed at room temperature overnight using the iBlot Western system (Thermofisher Scientific) containing primary antibodies, iBind wash solution, and horseradish peroxidase-conjugated secondary antibodies. Blots were developed using Clarity Western ECL (Bio-RAD, Irvine, CA, USA), and images were captured using a Fusion FX imaging machine (Vilber, Marne-la-Vallée, France). All antibodies were purchased from commercial sources and included the following: anti-HSF1 (catalogue number SPA-901), anti-HSP27 (catalogue number SPA-800), and anti-HSP90α (catalogue number SPA-835) antibodies from Enzo Life Sciences (Farmingdale, NY, USA); anti-HSP105/110 (catalogue number Sc-6241) antibody from Santa Cruz Biotechnology; anti-Ha-Ras (catalogue number 05-775) antibody from Merck Millipore (Darmstadt, Germany); anti-actin (catalogue number MS-1295-P0) and anti-HSP70i (catalogue number MS-482-P0) antibodies from Thermofisher Scientific; anti-p53 (catalogue number 51-9002046) from BD Pharmingen (San Diego, CA, USA); anti-pHER2/ErbB2 (catalogue number Ab53290), anti-pAkt (catalogue number MISCGENLAB), anti-pEGFR (catalogue number SCZSC-12351), and anti-total EGFR (catalogue number SCZSC-120) from Abcam (Boston, MA, USA); anti-total Akt (catalogue number 9272) and anti-HER2/ErbB2 (catalogue number Ab134182) from Cell Signaling (Beverly, MA, USA); and anti-EGR4 (catalogue number SC133540) from Santa Cruz.

### 2.6. RNA Extraction and Quantification

RNA was extracted using a Qiagen RNA extraction kit (Qiagen, Germantown, MD, USA) according to the manufacturer’s instructions. On-column DNA digestion was performed by adding 80 µL of DNase I in HDD buffer (Qiagen) onto the column and incubated at room temperature for 15 min. Following extraction, RNA was eluted in 40 µL of RNase free water, quantified using a NanoDrop 2000 machine (Thermofisher Scientificand then stored at −80 °C for subsequent use.

### 2.7. cDNA Synthesis

cDNA was synthesised using a SensiFAST cDNA synthesis kit (Bioline, Meridian Bioscience, Memphis, TN, USA) according to the manufacturer’s instructions. The concentration of all RNA samples was normalized prior to cDNA synthesis, and 80 ng of extracted RNA was combined with the Bioline reagents. After the cDNA synthesis was completed, the synthesized cDNA was diluted to a final concentration of 1 ng/µL for subsequent use in all qPCR reactions.

### 2.8. Quantitative PCR (qPCR) Analysis

All qPCR primers were designed using the NCBI primer designing tool (http://www.ncbi.nlm.nih.gov/tools/primer-blast/ accessed on 11 January 2017) and checked using Vector NTI software (Vector NTI Advance v11.5.2, Life Technologies). Each qPCR reaction (10 µL) was set up with 4 ng cDNA, 5 µL PerfeCTa Sybr Green Supermix (Quanta Biosciences, Beverly, MA, USA), and 1 µM forward and reverse PCR primers. Reactions were run on a BioRad CFX96 Real-time PCR system (Bio-rad Laboratories). The cycling conditions comprised of an initial 90 s denaturation at 95 °C, then 50 cycles of 95 °C denaturation for 10 s, and 60 °C elongation for 30 s. The final PCR products were examined using a melting curve analysis, with temperatures increased at a rate of 0.5 °C/sec, from 65 °C to 95 °C, to check for the presence of contamination and primer dimers.

### 2.9. Cell Proliferation Assay Using xCELLigence System

xCELLigence experiments were performed using the Real-Time Cell Analyzer instrument according to manufacturers’ instructions (ACEA Biosciences, San Diego, CA, USA). The optimal cell seeding number was determined by initial titration experiments. After seeding 10,000 cells/well, plates were loaded onto the machine, and automated cell index readings were taken every 30 min for 120 h. Lyophilised compounds were reconstituted in double-distilled water, and cells were treated with the compounds approximately 24 h after seeding, when the cells were in the log growth phase. For each assay, cells were treated with fresh media as control or with different concentrations of reconstituted drugs/compounds: lapatinib (0.0125, 0.025, 0.05, 0.1, 0.125, 0.2, 0.25, 0.5, 1, 2 μM), erlotinib (0.125, 0.2, 0.25, 0.5 μM), gefitinib (0.125, 0.2, 0.25, 0.5 μM), AUY922(12.5, 25, 50, 100, 200 nM), and sulforaphane (1.25, 2.5, 5 μM). At the completion of the experiment, the cell index value was calculated for each sample using the RTCA Software Package 1.2.

### 2.10. RNAseq Analysis

The dataset used in this study was extracted from the publicly available TCGA data set of mammary adenocarcinoma, downloaded from the Genomic data commons legacy archive (https://portal.gdc.cancer.gov/legacy-archive accessed on 15 July 2019). Level 3 standardized exon expression levels from 1097 tumor samples and 114 normal breast tissue samples were available on the RNASeqV2 platform. At the same time, exon annotations for the RNAseqV2 TCGA-Data were obtained from the TCGA website (https://tcgadata.nci.nih.gov/docs/GAF/GAF.hg19.June2011.bundle/outputs/TCGA.hg19.June2011.gaf accessed on 15 July 2019).

Available clinical information corresponding to 1085 patients was obtained and updated with the latest follow-up available. For further information about biospecimen collection, processing, quality control, and biomarker assessment, please refer to [28] or the TCGA website (http://cancergenome.nih.gov accessed on 15 July 2019).

### 2.11. Intrinsic Subtype Classification of TCGA Samples

The expression levels of 50 different genes from each sample were used to carry out the intrinsic subtype classification of tumours [29], which was performed using the “Bioclassifier” package (https://genome.unc.edu/pubsup/breastGEO/PAM50.zip accessed on 15 October 2016). To perform this task, the normalized expression profile (normalized RNA_SeqV2 RSEM) of the 50 specific genes was used. Many of these genes are strongly related with breast cancer behaviour and include ER1, ERBB2, PGR, and MKI67, among others. The expression values for each gene were standardized using the log2 expression levels and the median expression value of a subset of samples (50% estrogen receptor-positive and 50% estrogen receptor-negative population). Once the samples were classified, a principal component analysis, hierarchical cluster evaluations, and a sample-to-centroid correlation analysis were performed to assess the quality and validity of the classification (Appendix A available upon request).

### 2.12. Differential Exon Expression Analysis

To evaluate differentially expressed exons (DEEs), the EdgeR algorithm was used [30]. Briefly, this method uses a negative binomial generalized linear model and moderates the dispersion estimate for each gene toward a local estimate from genes with a similar expression strength. Exons with zero counts on all samples were filtered out. The trend dispersion was estimated with the empirical Bayes method provided with the “Bioclassifier” package (estimateTrenDisp function). A quasi-likelihood negative binomial generalized log-linear model was used to fit the count data (glmQLFit function). Once dispersion estimates were obtained and the negative binomial model was fitted, hypothesis testing was performed to determine the differential expression of exons using the empirical Bayes quasi-likelihood F-test (glmQLFTest function). The samples were grouped as ‘normal/not cancerous’ for the breast tissue samples and as ‘tumour-intrinsic molecular subtypes’ for the tumour samples. Log2 Fold change values were obtained and associated with *p*-values and False Discovery Rate values (FDR) by using the Benjamini and Hochberg method [31]. To assess differential exon usage, the log2-fold-change of each exon corresponding to the *EGR4* gene was compared to the log2-fold-change of the entire gene (diffSpliceDGE function).

### 2.13. TMA Construction

A tissue microarray (TMA) block containing breast tumour tissue and adjacent normal breast tissue from each patient was constructed by the Victorian Cancer Biobank, as described previously [32]. Two tumour cores and one normal breast tissue core from each patient were embedded in the TMA block. A total of 30 patients (operated on between 2002 and 2014), with a median age of 55 (range 27–91) years, were incorporated into the TMA block for evaluation.

### 2.14. Immunohistochemistry

TMA slides were deparaffinized in xylene, rehydrated in graded ethanol; then, antigen retrieval was performed by placing slides in a coplin jar with 10 mM citrate solution (pH 6.0), followed by incubation in an autoclave heated to 121 °C for 10 min. Endogenous peroxidase activity was blocked by incubating the slides in 0.3% H_2_O_2_ for 15 min at room temperature, and the slides were then placed in blocking solution containing 10% goat serum and 1% BSA for 2 h at room temperature. After blocking, the tissue specimens were incubated with either anti-HSF1 (Santa Cruz catalogue number SPA-901), anti-EGR4 (Santa Cruz catalogue number SC133540), or anti-HER2/ErbB2 (Abcam catalogue number Ab53290) overnight at 4 °C. After washing, the tissues were treated with a goat anti-rabbit biotin-conjugated secondary antibody (Abcam catalogue number Ab6720) for 1 h at room temperature. After washing the tissue sections, the localisation of antibody binding was made visible by using a 3,3-diaminobenzidine enhanced liquid substrate kit (Sigma Aldrich) according to the manufacturer’s instructions and by counterstaining the slides with Modified Harris Haemotoxylin (Thermofisher Scientific).

### 2.15. Histopathologist Grading of Staining Intensity

Tissue sections on the TMA slides were analysed in conjunction with histopathologist Dr Chris Dow. Tissue samples were examined under an Olympus BX53 microscope (Olympus) with multi-head attachments to determine the intensity of the brown DAB staining of nuclei for both EGR4-S and cell membranes for HER2. Each tissue section was subjectively classified into one of four categories depending on the intensity of the cellular staining: negative, weak, moderate, or strong. HER2 staining was specifically scored according to standard ASCO/CAP 2013 criteria for routine breast cancer HER2 immunohistochemistry reporting at the time [33].

### 2.16. Statistical Analysis

The cell number for each cell biology assay was normalised to its own arbitrary starting point. Normalised experimental data were then pooled for analysis, with the results presented as means ± standard error of the mean. Raw data are presented where possible; otherwise, median and interquartile range are presented. The Kruskal–Wallis test followed by Dunn’s test were used to determine whether the treatment groups were statistically significant compared with control. Comparisons between treatment groups over time were performed using the rank sum test separately at each time point, with adjustments for multiple comparisons using the Benjamini–Hochberg procedure.

Power calculations were performed for clinical sample numbers. Estimates for sample size calculations were derived from pilot data. To detect a difference in percentage of discordant pairs of 59% vs. 1%, with 80% power and 1% significance level (assuming a 30% drop out rate), a total of 26 samples were required. A total of 30 samples were obtained to retain 80% power even if a smaller difference was observed. McNemar’s test (paired proportions) was used to determine differences in protein expression between cancer and normal tissue. GraphPad Prism (V8) software was used for statistical analysis and generating figures. Significance in all tests is represented as either * *p* < 0.05, ** *p* < 0.01, or *** *p* < 0.001.

## 3. Results

### 3.1. EGR4 Expression Profile Is Affected by HSF1 in Breast Cell Lines

We previously generated MCF10A and ‘H-Ras^v12^ transformed’ MCF10A cells that constitutively express either a recombinant wild-type HSF1 (HSF1WT) or a constitutively active mutant of HSF1 (HSF1ΔRDT) in order to investigate the effects of HSF1 expression on oncogenicity in mammary epithelial cells [21]. Using these oncogenically transformed mammary cells, we initially assessed their glycolytic capacity with and without HSF1 activation since metabolic adaptations such as altered nutrient uptake and intracellular metabolism are emerging hallmarks of cellular transformation [34]. Results revealed that oncogenically transformed cells (Ras GFP) were significantly more glycolytic than non-transformed cells (Cherry GFP, *** *p* < 0.001) indicative of greater metabolic activity and lactic acid production (Appendix A). In comparison, upregulating HSF1 in the oncogenically transformed cells (Ras HSF1ΔRDT) resulted in significantly reduced glycolysis (** *p* < 0.01).

A microarray analysis was performed on the different cell populations to investigate the influence of HSF1 on transcript profiles in both transformed and non-transformed human mammary epithelial cells (unpublished data). This analysis identified *EGR4* as the one gene differentially regulated by HSF1 between transformed and non-transformed cells. Western blot analysis of cell lysates derived from H-Ras^v12^ transformed and non-transformed MCF10A cells revealed that the expression of EGR4 was increased in transformed (H-Ras^v12^) mammary cells as compared to controls (mCherry GFP Ctrl) (Figure 1A). The stable expression of HSF1 (HSF1WT or HSF1∆RDT) had a divergent effect on EGR4 protein levels, where HSF1 increased EGR4 levels in non-transformed MCF10A cells and yet decreased EGR4 levels in the H-Ras^v12^ transformed cells (Figure 1A).

We next examined EGR4 protein levels in a panel of human breast cancer cell lines by Western blot (Figure 1B). The results showed that EGR4 was expressed in most of the breast cancer cell lines from different sub-types. Consistent with our earlier finding that EGR4 expression is upregulated in Ras-transformed mammary cells (Figure 1A), EGR4 was highly expressed in cancer cells from the HER2+ subtype, which typically exhibit Ras pathway activation [35]. All cell types in the panel that showed high levels of EGR4 also exhibited a lower expression of HSF1, suggesting an inverse relationship between the two factors in transformed cells (Figure 1B). This pattern was also consistent with the previously observed lower levels of EGR4 in Ras-transformed MCF10A cells with HSF1 overexpression (Figure 1A). Notably, the EGR4 protein detected in these experiments was not the predicted size (expected to be more than 60 kDa) but rather a shortened version (~51 kDa), which we describe herein as “EGR4-S” (Figure 1C).

### 3.2. An EGR4 Splice Variant Derived from Exon 2 Is Expressed in Breast Cancer Cells

Inspection of the human *EGR4* consensus gene sequence showed that it has a two-exon structure, and that EGR4-S is most likely derived from a splice variant involving exon 2 that contains the DNA binding domain for its transcription factor activity (Figure 2A and Appendix A). To further examine the prevalence of the EGR4-S splice variant in different breast cancer contexts, we examined RNA-seq data across 56 different breast cancer cell lines available through the CCLE [36] (Figure 2B). The majority of the cell lines included in the dataset had detectable levels of *EGR4* mRNA (*n* = 41 of 56). Only exon 2 was expressed in the MCF10A sample of the CCLE dataset (Figure 2B), which is consistent with the observed shorter variant detected by Western blotting in Figure 1A. Of the 56 cell lines, 28 were found to only express exon 2, 13 expressed both exons 1 and 2, two cell lines expressed exon 1 alone, and 13 had no detectable expression of *EGR4*. These data, together with the observed ~51 kDa protein observed in Figure 1, indicate that the observed EGR4-S variant may be the product of exon 2 with the exclusion of exon 1. In cell lines where both exons were detected, the expression of exon 2 was much higher than that of exon 1 in all cases. EGR4 could also be readily detected by Western blot analysis of tumour lysates derived from HER2+ patient tumours (Figure 2C). Worthy of note is the fact that the EGR4-S variant was only detected in the HER2+ tumour tissue and not in any of the patient-matched normal tissues examined (Figure 2C). The higher molecular weight form of EGR4 was, however, more readily observed across the normal tissue samples (Figure 2C). From the 20 matched patient samples analysed (Figure 2C), EGR4-S was detected in 80% of all HER2+ breast cancer tissues and none of the matched control tissues (with a difference in proportion between cancer and normal breast tissue of 0.80 (95% CI 0.62, 0.98), *p* < 0.001).

### 3.3. Expression of EGR4 Exon 2 Splice Variant in Human Clinical Samples

We next conducted a qPCR quantification of *EGR4-S* mRNA levels with mRNA isolated from breast tumour biopsies taken from three patients (Figure 3A), using primers that bound to either exon 2 or primers that spanned the exon1/exon 2 boundary (‘exon 1′). Across the samples examined, exon 2 primers amplified a PCR product, with products from the exon 1 primers bordering on being undetectable under the parameters used (Figure 3A). Analysis of *EGR4* exon expression across the TCGA BRCA RNA-seq dataset (*n* = 1085) found *EGR4* exon 2 to be expressed at relatively comparable levels across breast tumours of different subtypes, with the median expression of exon 2 being higher than exon 1 for each subtype. Additionally, exon 2 was almost exclusively expressed within the HER2 subtype (Figure 3B). Following this, we examined whether EGR4 could be readily detected by immunohistochemical (IHC) staining in human (HER2+) breast tumours (Figure 3C). IHC staining for HER2, HSF1, and EGR4 in tumour punch biopsies revealed the positive nuclear staining of HSF1 and EGR4 in tumours presenting with positive membrane staining for HER2 across 15 tumour samples (Figure 3C). When the observed specific expression of EGR4-S in HER2+ cells is considered with higher EGR4-S protein levels in the HER2+ breast cancer cell lines (Figure 1), the data suggest the existence of a distinct mechanism of EGR4 regulation within HER2+ human mammary tumour cells that is not active in normal tissue or in tumour tissue of other breast cancer subtypes.

### 3.4. Inverse Relationship between EGR4 and HSF1 Expression

As our earlier studies identified the modulation of EGR4 expression by HSF1 in a transformation-dependent manner, we further investigated the relationship between HSF1 and EGR4-S expression. Modifying MCF7 and T47D cells (identified in previous experiments as having relatively lower levels of EGR4-S at 51 kDa—Figure 1B) with the stable over-expression of HSF1 (HSF1 WT and HSF1 ∆RDT) led to further reduced EGR4 levels (Figure 4A). In the HER2+ cell lines, MDA-361 and SKBR3, the ectopic overexpression of HSF1 also led to a lower expression of EGR4-S (Figure 4A). Consistent with the observed inverse relationship between HSF1 and EGR4-S, the knockdown of *HSF1* in Hs578T basal breast cancer cells, which have high endogenous HSF1 protein levels, was associated with the higher expression of EGR4-S (Figure 4B). In further agreement with an inverse relationship between HSF1 activity and EGR4-S expression, the HSF1-activating compounds AUY922 and sulforaphane (SFN) also reduced EGR4-S protein levels in cultured SKBR3 cells after 24 h of treatment (Figure 4C,D).

### 3.5. Inhibiting HER Signalling Pathway Reduces EGR4 Expression

Given our observation that EGR4-S levels were typically higher in HER2+ cancer cells, we next examined the response of EGR4-S to treatment with clinically approved cancer therapeutics that target the HER signalling pathway. The treatment of HER2+ cells with tyrosine kinase inhibitors (lapatinib, gefitinib, and erlotinib) led to reduced levels of EGR4-S (Figure 5A,B). The reduction in the 51 kDa EGR4-S observed with the increase of drug concentrations mirrored the reduction observed in other upstream components of the HER pathway. This loss in EGR4-S expression with drug treatment was not transient but rather maintained over time (Figure 5B). Results from similar experiments on EGFR (HER1-activated) breast cancer cells such as MDA-468 showed that the effect of drug treatment on EGR4-S levels within this context was still present but not as robust as that observed in HER2+ cell lines (Figure 5C).

### 3.6. EGR4-S Reduction Caused by HER Pathway Inhibition Is Augmented by HSF1

Following the observation that EGR4-S expression responds to HER pathway targeted drug treatment, we investigated whether this phenomenon could be influenced by modifying the cellular expression of HSF1. Lapatinib treatment reduced EGR4-S expression in a dose-dependent manner, and this reduction was more pronounced with elevated HSF1 (Figure 6A). When HSF1 levels were experimentally lowered in the cells by the expression of HSF1-targeted shRNAs (Figure 6B), EGR4-S expression did not respond as readily to equivalent lapatinib doses. We also cultured the cell line in increasing concentrations of lapatinib over the course of seven weeks to investigate whether EGR4-S expression would be affected during prolonged treatment with lapatinib. After 3 weeks of treatment, reduced levels of EGR4-S could be observed compared to untreated control samples (Figure 6C). However, with prolonged lapatinib treatment, EGR4-S expression was found to be unchanged and expressed at a level comparable to that of untreated control cells. A comparison of HER2 and EGR4-S protein expression on biopsies taken from eight different breast cancer patients diagnosed with HER2+ tumours (Figure 6D) showed that no EGR4-S protein could be detected in patients that received pre-biopsy treatment with drugs to suppress the HER2 pathway. We therefore conclude that EGR4-S expression is dependent upon HER2 pathway activation, and that EGR4-S may have some value as a biomarker for the inhibition of the HER2 pathway.

### 3.7. EGR4-S Knockdown Reduces Breast Cancer Cell Growth

Since we had observed the inverse association between HSF1 and EGR4-S in HER2+ cancer cells across several experimental conditions, we decided to examine the effect of directly manipulating EGR4 and HSF1 levels on breast cancer cells. Increasing HSF1 in SKBR3 (HER2+) cells resulted in a significantly lower proliferation rate over time as compared to control cells (Appendix A, * *p* < 0.05, ** *p* < 0.01). Knocking down EGR4 (Figure 7A) in SKBR3 cells also resulted in a reduced growth rate over time. In addition, results from growth and migration assays (Appendix A) demonstrated that elevated HSF1 resulted in a more disorganised growth pattern in 3D culture, and that cells exhibited significantly more migratory behaviour with higher HSF1 (WT *** *p* < 0.001, RDT *** *p* < 0.001), both characteristics of a more metastatic cancer phenotype.

Based on these collective findings, which demonstrate that the activation of the HER signalling pathway is associated with EGR4-S expression, that EGR4-S expression is also associated with cancer cell proliferation and drug responsiveness, and that EGR4-S expression is influenced by HSF1, we developed a schematic diagram to summarise the various interactions observed across our experimental results (Figure 7B).

## 4. Discussion

In this work, we have identified a novel splice variant of this stem cell transcription factor (EGR4-S) that is yet to be characterised in a cancer context. Interestingly, the protein detected in our experiments was not the predicted full-length form of EGR4 (expected to be more than 60 kDa) but rather a shortened version (~51 kDa) that is responsive to HSF1 and HER pathway signalling. Inspection of the *EGR4* gene sequence showed that exon 2 contains three DNA binding domains that enable the molecule to function as a transcription factor, and our results suggest that this novel splice variant is most likely derived from exon 2.

*EGR4* was first characterised in the central nervous system [37,38], and research has since shown that this neural expression is transient in the developing vertebrate brain [39]. Roberts and colleagues [40] showed that neurons within the brain can live for the entire lifespan of the host organism, and this lifespan extends when the neurons are transferred to a new host [41]. Further to this, EGR4 expression has also been isolated in human testes, where it is localised within stem cells that survive across the lifespan to produce spermatozoa [42]. Di Persio et al. [43] also demonstrated that EGR4 is upregulated in the most undifferentiated spermatogonial cells, and that it is an upstream regulator of several transcription factors involved in cell proliferation and differentiation. Our research shows that EGR4-S was upregulated with the oncogenic transformation of mammary cells in vitro and was detected in cancer tissue but not in patient-matched normal tissue, suggesting that the expression of this splice variant is a characteristic of the cancer cell. This is supported by the patient database analysis conducted by Fei et al. [24], who suggested that *EGR4* may be a potential oncogene in breast cancer after showing that the upregulation of EGR4 is correlated with breast cancer grade.

In other tumour contexts, EGR4 has also been connected to advanced tumourigenesis. For example, Matsuo et al. [22] previously identified EGR4 as having a role in cell proliferation and the bone metastasis of small cell lung cancer through its direct interaction with the promoter regions of target genes such as SAMD5, RAB15, SYNPO, and DLX5. Additionally, research by He et al. [23] demonstrated that the 3′ region of *EGR4* (located on exon 2) can interact with the promoter region of the ZNF205-HS1 gene in order to promote non-small cell lung cancer cell growth in vitro and in vivo. Given that EGR4 has been characterised in neurons, embryonic cells as well as stem cells that have longer lifespans than other body cells, it is possible that EGR4-S expression plays a role in cancer cells developing the characteristic of prolonged survival or immortalisation.

As with stem cells, established cancer cells have endless proliferative potential enabled by the activation of oncogenic pathways such as the HER pathway [44]. Here, we have shown for the first time that EGR4-S is highly expressed in HER2+ breast cancer cell lines and clinical tumour samples, and that EGR4-S levels are subject to HER2 signalling activity. This is evidenced by the fact that EGR4-S expression could be increased or decreased in direct relation to altered components of the signalling pathway such as Ras, pHER2, and pAKT levels or the chemical inhibition of HER2. Furthermore, knocking down *EGR4-S* was found to influence the growth properties of HER2+ SKBR3 cells by reducing cell growth over time. This result is consistent with previous findings which show that the knockdown of *EGR4* significantly suppresses cancer cell growth rates in small cell lung cancer cells [22] and cholangiosarcoma cells [45], reduces cell viability and proliferation in non-small cell lung cancer cells, and reduces tumour growth using an in vivo mouse model [23].

### 4.1. EGR4-S as a Biomarker

Historically, targeted therapies for EGFR/HER positive cancer treatment have focussed on upstream components of the HER signalling pathway [46,47,48,49]. TKI drugs such as lapatinib are approved for clinical use in treating HER-overexpressing cancers [47]. Our results show that EGR4-S may have value as a pharmacodynamic biomarker for TKI drug treatment in HER2+ cancer cells. Our findings here also identified EGR4-S expression to be specific to tumourous tissues and was not identified in patient-matched normal tissue, although it was expressed in spontaneously immortalized MCF10A cells. These early findings warrant further examination of EGR4-S expression in a larger dataset of tumour tissue with normal matched controls. Based on the data presented here, such a study has potential to identify EGR4-S as a tumour-specific factor that may help distinguish tumour tissue from normal tissue at the molecular level.

Unfortunately, resistance to targeted therapies develops in many patients [50,51,52]. In such cases, strategies such as intensifying the treatment by combining HER2-targeted therapies together or extending the duration of treatment are often employed [53]. Examining EGR4-S expression in cancer cells that underwent sustained (7-week) treatment with high concentrations of lapatinib revealed that, as time progressed, EGR4-S was no longer responsive despite further drug treatment. Goutsouliak and colleagues [53] suggest that new predictive biomarkers for HER-targeted therapies are currently needed as not all patients derive enough benefit from the treatment to outweigh the negatives of toxicity and cost. In light of our findings that EGR4-S decreases with HER2 inhibition in cells sensitive to HER2 inhibitors, our studies indicate that EGR4-S has potential value as a biomarker for tracking TKI drug responsiveness in HER2+ cancer cells. Furthermore, the identification of EGR4-S as a regulator of cell growth in cancer cells and a downstream target of the HER pathway, combined with the fact that our results suggest its unique expression by cancer tissue, could allow for the development of another targeted therapy in the future.

### 4.2. Cellular Stress and EGR4-S

Our data presented here, along with observations from our previously published cell model [21] showing that increased HSF1 in oncogenically transformed breast cancer cells promotes a more metastatic phenotype, are consistent with research performed by others who have also shown that elevated cellular stress can accelerate cancer progression [54,55,56,57], and that metastatic tumours show a global increase in stress response compared to normal tissues [58]. HSF1 itself has long been reported to play critical roles in cancer progression, metastasis, and drug resistance [20]. Compounds that activate a cellular stress response such as HSP90 inhibitors (AUY922, 17AAG, geldanamycin) have limited effectiveness in cancer and can even promote metastasis [27,59]. Our studies herein found that chemical inducers of proteotoxic stress affect *EGR4-S* expression. These data support our findings on cultured cancer cells with the ectopic overexpression of HSF1 since these chemicals also activate HSF1. SFN, for example, has been shown to initiate a heat shock response in mammalian cells, and its mechanism for activating stress and HSF1 is different to that of HSP90 inhibitors [60,61,62].

Interestingly, research by Yallowtiz and colleagues [63] showed that lapatinib-resistant breast cancer cells have chronically activated HSF1 and heat shock proteins, but that these cells can be sensitised to treatment using HSF1 inhibitors. Bioinformatic analysis of patient databases conducted by Fei et al. [24] showed that EGR4 levels are significantly lower in triple negative breast cancer patients, and that EGR4 is downregulated with metastatic breast cancer (i.e., when nodule status was positive). Taken together with our observation of the inverse relationship between HSF1 and EGR4-S expression, the relationship to metastasis would be interesting to investigate further. It is possible that there are two alternative cell lineage pathways that a cancer cell can follow—increased growth and proliferation, or enhanced migration and metastasis. Heightened molecular stress in the cell could reduce the cancer cell growth, but in doing so, it could trigger the cell to follow a different pathway to cancer progression and metastasis. In this hypothetical scenario, a stress-driven reduction in EGR4-S would result in a worse prognosis for long-term treatment.

### 4.3. Limitations

The successful cloning and sequencing of *EGR4-S* cDNA would provide a more precise definition of the EGR4-S isoform as well as additional insight into its transcription/translation regulation and other biologically relevant structural features.

Several efforts were made to clone the full-length *EGR4* from SKBR3 cells and other breast cancer cell lines. cDNAs were generated from these cells; however, all primers were designed using the published full-length *EGR4* sequence. Our inability to clone the full-length *EGR4* from various breast cancer cell lines, combined with the successful detection of amplicons from exon-2 (Figure 3A), did support our hypothesis for the existence of a shortened version of the protein (EGR4-S). However, the difficulties we encountered in locating a starting point for the transcription of *EGR4-S* mRNA led us to use the published full-length *EGR4* sequence to create predictions for EGR4-S, as shown in Figure 2A and Appendix A. Future studies that identify the *EGR4-S* coding region will provide an important advancement in the further characterisation of the *EGR4-S* isoform.

## 5. Conclusions

Until now, the shorter splice variant of EGR4 that is present in cancer cells (EGR4-S) has been unknown. The full significance of this finding will likely become apparent with the further characterisation of EGR4 and EGR4-S functions in cancer. We have also demonstrated that there is an inverse relationship between HSF1 activity and the expression of the novel EGR4-S molecule in cancer. These findings suggest that relative levels of HSF1 and EGR4-S found together in the tumour tissue may provide a valuable indicator or a responsiveness to HER-targeted treatments and/or other tumour properties. The potential of these possibilities will continue to emerge with the further characterization of EGR4-S biology.

## 6. Patents

See ref. [64].

## Figures and Tables

**Figure 1 cancers-14-01567-f001:**
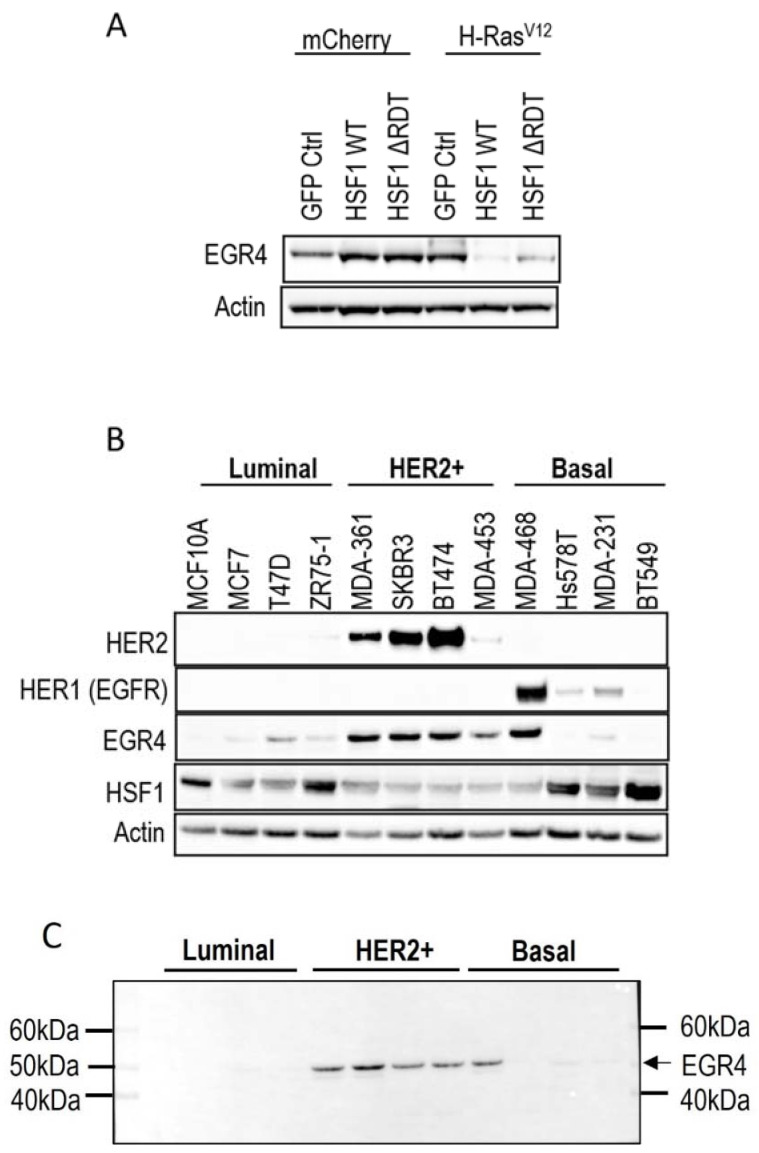
Association of EGR4 with HSF1 and HER2/HER1 expression in breast cell lines. (**A**) EGR4 protein expression in non-transformed breast cells (mCherry) and oncogenically transformed breast cells (H-Ras^V12^) with low (GFP Ctrl) and high HSF1 (HSF1 WT, HSF1 ΔRDT). (**B**) Protein expression in a panel of breast cancer cell lines of different cancer sub-types (Luminal, HER2+, Basal). (**C**) Merged light/chemiluminescent image showing molecular ladder (left and right) and EGR4 protein bands detected around ~51 kDa.

**Figure 2 cancers-14-01567-f002:**
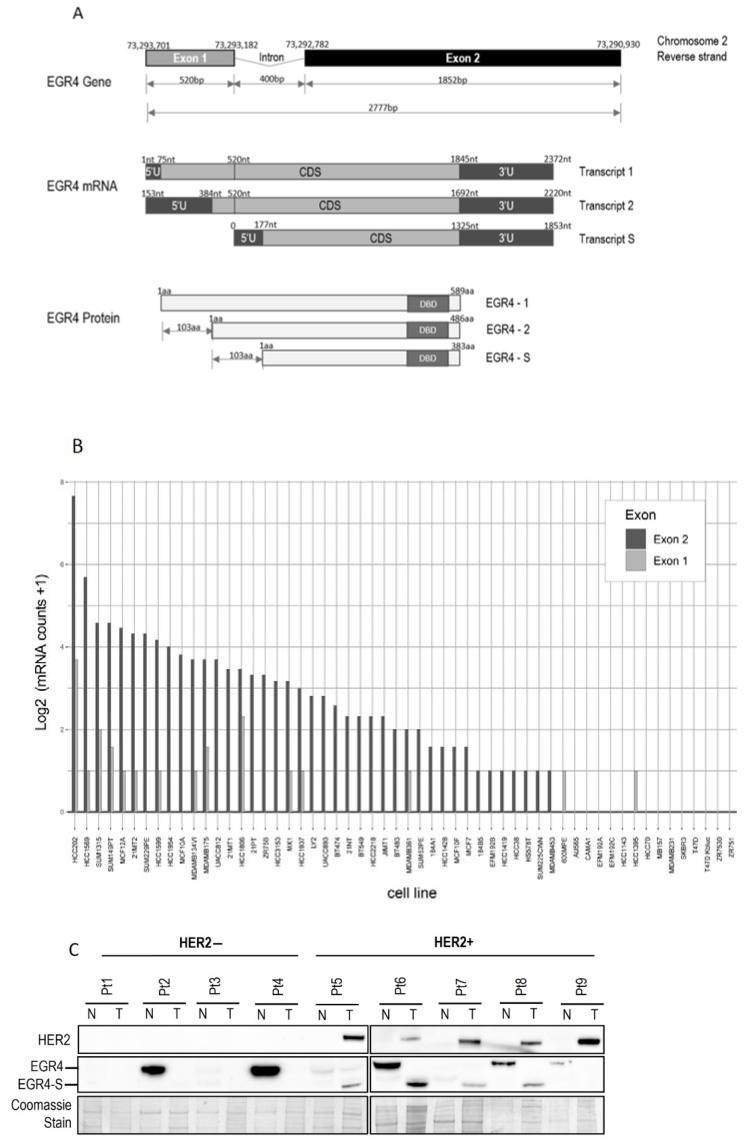
Structure of the *EGR4* gene and splice variant expression in cells and tumour tissue. (**A**) Schematic diagram of the Human *EGR4* gene, located on Chromosome 2, showing its 2 exon structure (**top**), the 3 possible mRNA transcripts (**middle**), and 3 hypothetical protein isoforms arising from these different transcripts (**bottom**). Based on: https://www.ensembl.info/known-bugs/ensembl-100/ (accessed on 15 August 2017). (**B**) RNAseq analysis for *EGR4* exon expression in 56 different breast cancer cell lines. Reads were normalised to exon size by scaling to base-level coverage. (**C**) Example of Western blot results for HER2 and EGR4 expression in normal (N) and breast tumour (T) biopsies from HER2− (pt 1–4) and HER2+ (pt 5–9) patients.

**Figure 3 cancers-14-01567-f003:**
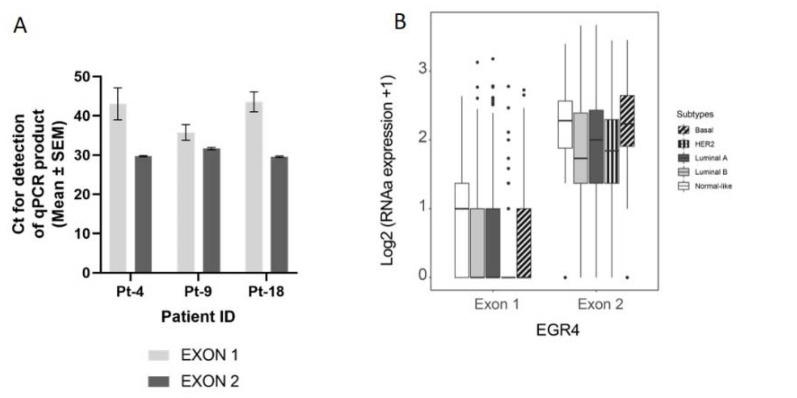
Confirmation of the EGR4-S splice variant in cell lines and clinical samples. (**A**) Results from qPCR on three breast tumour patient biopsies showing the amplification cycle for a qPCR product (mean ± SEM) detected using primers binding in exon 1 vs. primers binding in exon 2 for the same tumour sample. (**B**) Analysis of TCGA BRCA RNA-seq dataset (*n* = 1085) showing distribution and median proportional change for the expression of *EGR4* exon 1 and exon 2 mRNA in patients diagnosed with breast carcinoma (*n* = 1085). Expression of mRNA is separated according to breast cancer sub-type. (**C**) Representative punch biopsies from a HER2+ breast tumour at low magnification (left panels) and high magnification (right panels).

**Figure 4 cancers-14-01567-f004:**
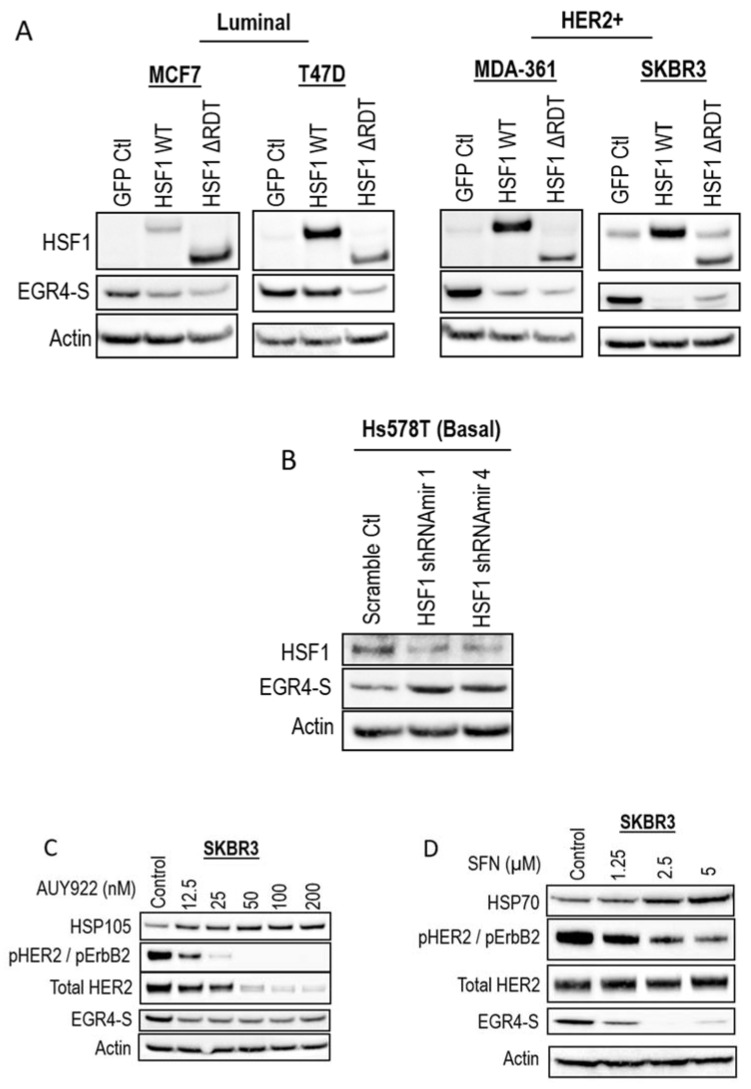
Inverse association between HSF1 and EGR4-S expression. Western blots for EGR4-S and HSF1 expression in (**A**) Luminal, HER2+, and (**B**) Basal breast cancer cells, with overexpression or shRNA-knockdown of HSF1, respectively. (**C**) Western blots of HER2+ cells treated with varying concentrations of the stress-inducing compound, AUY922, for 24 h and (**D**) sulforaphane (SFN) for 24 h.

**Figure 5 cancers-14-01567-f005:**
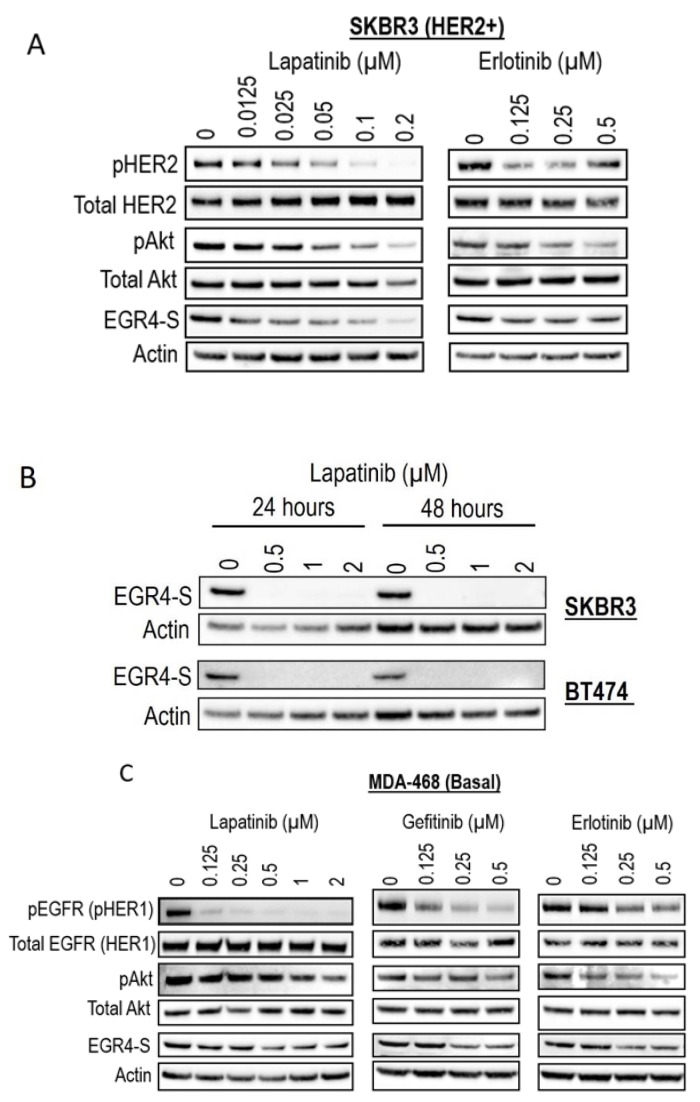
EGR4-S expression is regulated by HER- pathway targeted drug treatment. (**A**) Western blot analysis of HER2+ cell lysates treated for 24 h with increasing concentrations of TKI drugs (lapatinib, erlotinib) and (**B**) performed on indicated cell lysates treated over 24–48 h. (**C**) Western blot analysis of MDA-MB-468 breast cancer cell lysates treated with increasing concentrations of TKI drugs for 24 h (lapatinib, gefitinib, erlotinib).

**Figure 6 cancers-14-01567-f006:**
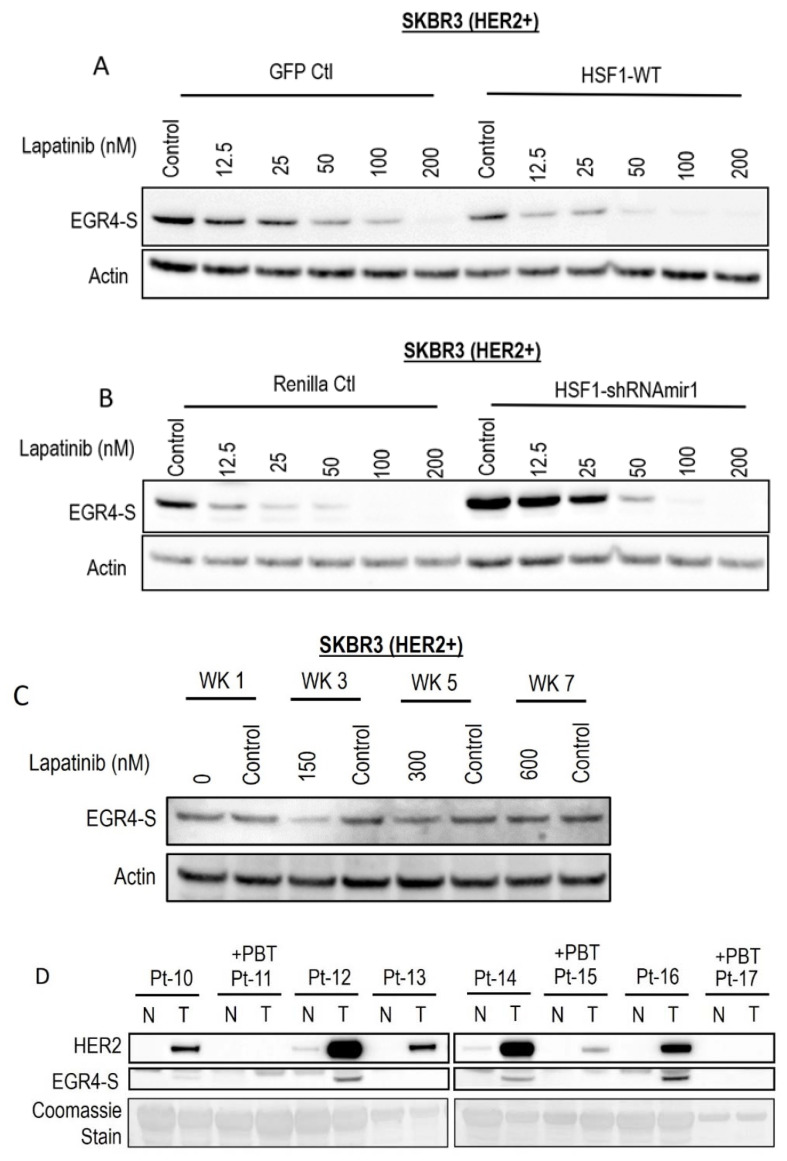
EGR4-S expression is responsive to HER-targeted treatment and sensitive to altered HSF1 activity. (**A**) Lapatinib treatment of HER2+ cells, with/without elevated HSF1 and (**B**) with/without HSF1 shRNA knockdown. (**C**) EGR4-S expression in cells grown in increasing concentrations of lapatinib over the course of 7 weeks exhibits some reduction visible at 3 weeks. (**D**) Comparison of HER2 and EGR4-S protein expression in tumour (T) tissue and adjacent normal (N) tissue from eight different breast cancer patients (Pt 1–8) diagnosed with HER2+ tumours. Pt 11, 15, and 17 received pre-biopsy treatment (+PBT) to suppress the HER2 pathway.

**Figure 7 cancers-14-01567-f007:**
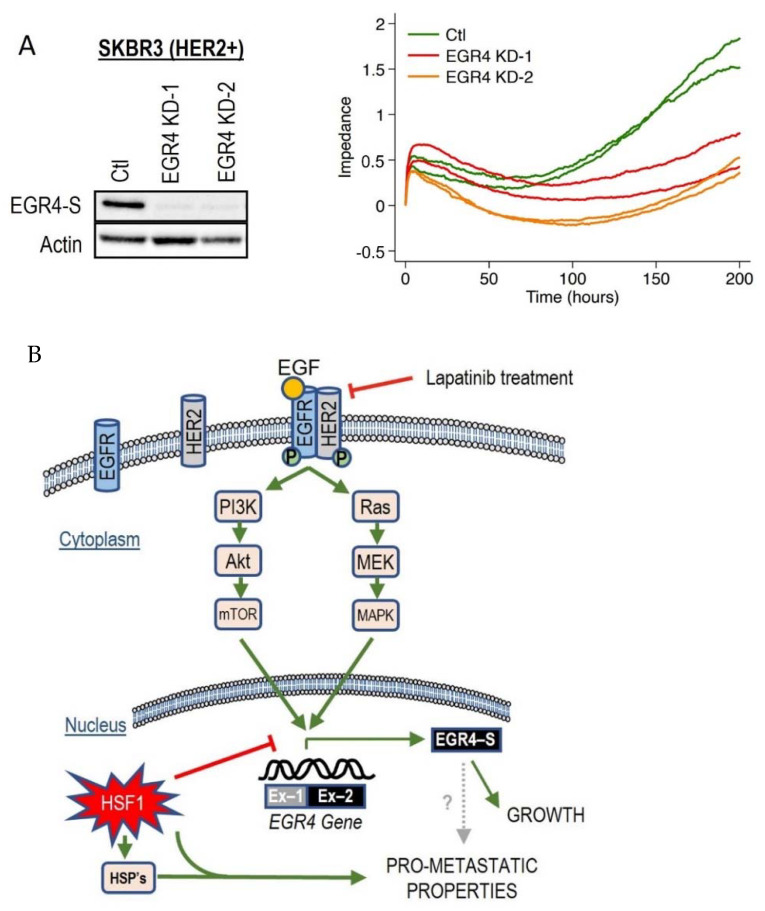
Effect of altered EGR4 expression on cancer cell growth. (**A**) xCELLigence Assay for HER2+ cell viability with/without *EGR4-S* knockdown (EGR4-KD) over time. (**B**) A schematic of the proposed EGFR(HER1)/HER2 signalling pathway and its relationship with the downstream transcription factor, EGR4-S.

## Data Availability

The dataset used in this study was extracted from the publicly available TCGA data set of mammary adenocarcinoma downloaded from Genomic data commons legacy archive (https://portal.gdc.cancer.gov/legacy-archive accessed on 15 July 2019). At the same time, exon annotations for the RNAseqV2 TCGA-Data were obtained from the TCGA website (https://tcgadata.nci.nih.gov/docs/GAF/GAF.hg19.June2011.bundle/outputs/TCGA.hg19.June2011.gaf accessed on 15 July 2019).

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
