# Peer review of "Regulation of a Novel Splice Variant of Early Growth Response 4 (EGR4-S) by HER+ Signalling and HSF1 in Breast Cancer"

_cancers, 2022, doi:10.3390/cancers14061567_

Round 1

Reviewer 1 Report

In this study, the authors identified a new, shortened version of EGR4 (EGR4-S) found in breast cancer but not detectable in normal breast tissue. It is regulated upregulated with high EGFR, HER2 and H-Ras expressing breast cell lines inhibited in response to HER inhibitors. The authors used multiple assays to test the potential function and proposed that it can be a potential cancer biomarker. The work was rather extensive, and the evidence were supportive. Further characterization is warranted.

Major comments:

A major weakness of the study is, while many efforts were made in functional test for EGR4-S, the authors didn’t provide the actual sequence information for EGR4-S, instead of relying in database information to guess what it looks alike. In my mind, this is a big mistake. It can be easily achieved by RT-PCR/cloning/sequencing to generate a full-length EGR4-S cDNA and use the sequence to precisely determine EGR4-S’s structural features. Without the sequences, the data from downstream hard work can’t be certain.

The description for the relationships among HPS, HSF1, EGR4 and EGR4-S is also rather confusion. In Figure 3B, should the significance be between exon 1 and exon 2 rather than among cell-types? Gene name should be in italic.

Reviewer 2 Report

In this manuscript, Drake et al, identified a novel shorter variant EGR4-S of EGR4 and have demonstrated its role as a biomarker for HER2 pathway activation in human tumors which are known to be regulated by heat shock factor-1. The authors have used a combination of overexpression, knockdown systems and pharmacological interventions to illustrate that EGR4-S expression is regulated by inactivation of HER2 pathway and/ or overexpression of HSF1.  This study indeed contributes to the field and helps to identify a novel biomarker for predicting patient prognosis and development of resistance to targeted therapies. However, there are some major concerns with the manuscript that are enlisted below:

  • Introduction:
    • The authors have given too much background information about the cellular stress response and no information about EGR4. It seems that this is a follow up study to a previously published paper by the same group for evaluating role of HSF1 in oncogenecity in MCF10A cell lines. For reader understanding, it would be best if the authors summarize just the important findings of that paper in a paragraph and then introduce EGR4 which came up from the microarray as no microarray data is presented in this paper and the reader is left wondering as to from where did EGR4 came from. Therefore, please delete excessive information about the cellular stress pathway and introduce more about EGR4 in introduction rather talking about it in discussion.

  • Methods
    • Lines 122-123, were the stable cell lines established based on GFP or were they selected using Puromycin/ any other selection marker. Please provide the details of the vector constructs and sequences that were used to make the over-expression and knockdown systems. This needs to be updated in the method section. Also, please provide the catalog numbers of the different chemical/ pharmacological inhibitors used in the experiments.
    • Lines 131-141, the authors need to elaborate about the exact protocol used for the glycolysis assay. What were the concentrations of glucose, oligomycin and 2-DG used in the experiment? Also, ‘cell metabolism’ is a very broad term and in the current research about tumor metabolism also encompasses fatty acid metabolism, amino acid metabolism, etc. Essentially, the authors just evaluated the glycolytic status of the cells. Please rename the section accordingly.
    • Lines 208-212, please clarify whether the control was just fresh media or the media containing vehicles used for dissolving the drug? If the authors just used fresh media without the vehicle then it does not serve as the appropriate control for the experiments.
    • Lines 295-302, the authors state that student’s t-test was used for statistical analysis; however, for experiments with multiple groups one-way ANOVA with multiple comparisons should be employed. Please re-analyze the data from the experiments that have multiple groups by using ANOVA instead of t-test. Also, for cell proliferation data which has a time and treatment component a two-way ANOVA should be employed. Please also mention the software that was used to perform statistical analysis.

  • Results
    • Overall, the results section needs to have sub-headings which discuss each figure instead of just continuous streams of paragraphs to improve the reading experience for the manuscript.
    • For all the western blots, please provide the quantifications with the standard deviations to understand the extent of variability. Please also include the molecular weight markers for all the western blots.
    • Lines 316-318: The authors introduce the microarray data from which EGR4 was identified as a protein of interest but it is unclear if it was a top hit and also please provide a cross-reference to the data.
    • Figure 3A, were any healthy control samples also used? It is important to know the differences between healthy control vs. patient, since the authors claim that EGR4 is not expressed in healthy tissue but only in tumor tissue.
    • Fig 5A, C: What were the treatment duration for these experiments and also please include the rationale for selecting the tested dose ranges? Are they physiologically relevant i.e., similar to those achieved in patients using these drugs?
    • Lines 446-453, there is no mention of this chronic treatment experiment with lapatinib in the method section. This experimental design should be updated in the method section. Please detail the frequency at which the dose escalation was performed. Did the authors check the effect on EGR4-S when cells are exposed to sub-therapeutic concentrations of lapatinib for prolonged period of time (classical model for development of drug resistance)?

  • Discussion

The section needs improvement and needs to be shortened. There is too much information about EGR4 which should actually be part of the introduction. Please also include limitations of your study. 

Round 2

Reviewer 1 Report

The revision has addressed most of my questions and its quality has been substantially improved. I suggest to publish the revisino in Cancers.

Reviewer 2 Report

The authors have addressed all the comments. The manuscript is improved after incorporation of all the suggested changes.